# The Diagnosis and Treatment Approach for Oligo-Recurrent and Oligo-Progressive Renal Cell Carcinoma

**DOI:** 10.3390/cancers15245873

**Published:** 2023-12-17

**Authors:** Kensuke Bekku, Tatsushi Kawada, Takanori Sekito, Kasumi Yoshinaga, Yuki Maruyama, Tomoaki Yamanoi, Yusuke Tominaga, Takuya Sadahira, Satoshi Katayama, Takehiro Iwata, Shingo Nishimura, Kohei Edamura, Tomoko Kobayashi, Yasuyuki Kobayashi, Motoo Araki, Yuzuru Niibe

**Affiliations:** 1Department of Urology, Graduate School of Medicine, Dentistry and Pharmaceutical Sciences, Okayama University, Okayama 700-8558, Japan; p7ez8zvc@s.okayama-u.ac.jp (T.K.); pu1o0fqk@s.okayama-u.ac.jp (T.S.); pzxh8r1r@s.okayama-u.ac.jp (K.Y.); p1fa9dra@okayama-u.ac.jp (Y.M.); ppv86vc7@s.okayama-u.ac.jp (T.Y.); p3uq1s4o@s.okayama-u.ac.jp (Y.T.); pnrs4ypc@s.okayama-u.ac.jp (T.S.); pzgr6t3w@s.okayama-u.ac.jp (S.K.); pfty8bwj@s.okayama-u.ac.jp (T.I.); pm5n8mce@s.okayama-u.ac.jp (S.N.); edakohe@okayama-u.ac.jp (K.E.); pbsz7o59@s.okayama-u.ac.jp (T.K.); kobayasu@md.okayama-u.ac.jp (Y.K.); motoosh@md.okayama-u.ac.jp (M.A.); 2Department of Public Health, School of Medicine, Kurume University, Fukuoka 830-0011, Japan; joe-n@hkg.odn.ne.jp

**Keywords:** renal cell carcinoma, oligo-metastasis, oligo-recurrence, oligo-progression, metastasectomy, stereotactic body radiation therapy

## Abstract

**Simple Summary:**

Oligo-recurrence and oligo-progression in oncology refer to scenarios in which five or fewer metastases develop after the primary lesions and other metastases have been treated using radical surgery or controlled using systemic therapy. Such scenarios are now considered favorable indications for metastasis-directed treatment (MDT), where a curative approach is possible. The primary strategy for recurrence and metastases in renal cell carcinoma (RCC) is systemic therapy, which includes immune checkpoint inhibitors (ICIs) and vascular endothelial growth factor receptor tyrosine kinase inhibitors. However, MDTs such as stereotactic body radiation therapy, which involves larger doses administered over fewer fractions than conventional radiation therapy, have shown the potential to improve outcomes for selected patients with RCC experiencing oligo-recurrence and oligo-progression. Combining MDT with ICIs may yield better results owing to their synergistic effects. In this review, we provide an overview of the current evidence related to the management of oligo-recurrent and -progressive RCC.

**Abstract:**

One-third of renal cell carcinomas (RCCs) without metastases develop metastatic disease after extirpative surgery for the primary tumors. The majority of metastatic RCC cases, along with treated primary lesions, involve limited lesions termed “oligo-recurrent” disease. The role of metastasis-directed therapy (MDT), including stereotactic body radiation therapy (SBRT) and metastasectomy, in the treatment of oligo-recurrent RCC has evolved. Although the surgical resection of all lesions alone can have a curative intent, SBRT is a valuable treatment option, especially for patients concurrently receiving systemic therapy. Contemporary immune checkpoint inhibitor (ICI) combination therapies remain central to the management of metastatic RCC. However, one objective of MDT is to delay the initiation of systemic therapies, thereby sparing patients from potentially unnecessary burdens. Undertaking MDT for cases showing progression under systemic therapies, known as “oligo-progression”, can be complex in considering the treatment approach. Its efficacy may be diminished compared to patients with stable disease. SBRT combined with ICI can be a promising treatment for these cases because radiation therapy has been shown to affect the tumor microenvironment and areas beyond the irradiated sites. This may enhance the efficacy of ICIs, although their efficacy has only been demonstrated in clinical trials.

## 1. Introduction

Kidney cancer is the sixth and ninth most common malignancy in males and females, respectively [1]. Renal cell carcinoma (RCC) is the predominant histologic type, comprising 85% of all primary renal neoplasms [2,3]. At the time of diagnosis, approximately 20% of patients with RCC already have metastatic lesions [4]. Additionally, 30% of initially non-metastatic RCCs develop distant metastases after radical surgery [5,6]. The onset of metastases is linked to a poor prognosis, with a 5-year overall survival (OS) of approximately 15% [3,7]. Although the advent of novel immune checkpoint inhibitors (ICIs) and their combination therapies has improved patient outcomes, metastatic RCC remains one of the malignancies with the poorest prognosis [8]. However, certain patients with metastatic RCC exhibit a more favorable prognosis and can be grouped into risk categories based on an established risk model, such as the International Metastatic RCC Database Consortium (IMDC) risk stratification [9].

In 1995, Hellman and Weichselbaum introduced a transformative concept, oligo-metastasis, to our understanding of cancer metastasis and recurrence [10]. This refers to a limited number of metastatic lesions [11]. Evidence regarding the treatment of oligo-metastatic RCC has evolved [12]. These findings largely affirm that metastasis-directed treatments (MDT) such as stereotactic body radiation therapy (SBRT) and metastasectomy are beneficial [13,14,15]. A large cohort study indicated that the majority of initially metastatic RCC had a single metastatic site, suggesting that oligo-metastatic RCC was the most common metastatic RCC [16]. MDT is an effective treatment option for patients with a few metastatic lesions [12,17]. In contrast, metastatic RCC is diverse and spans a spectrum of disease severities [6,8]. Some patients experience disease progression even after undergoing appropriate local treatments. Although retrospective data consistently indicate that the complete resection or SBRT of solitary or limited lesions leads to a better prognosis [18,19], there is still debate about whether this is due to favorable tumor biology, the impact of MDT, or a combination of both. Therefore, identifying ideal candidates based on the cancer condition that predicts disease progression is paramount. Moreover, determining the appropriate timing and selecting the best treatment modality is crucial for managing oligo-metastatic RCC, with the overarching goal of enhancing patient outcomes.

In this review, we provide an overview of the distinct characteristics of oligo-metastatic RCC and discuss optimal treatment approaches for these patients.

## 2. Metastatic Patterns in RCC

Several population-based studies have investigated metastatic patterns in RCC. However, data on the pattern of metastatic spread following radical surgery for localized disease remains limited. One study based on a Swedish cohort of 3107 patients with RCC revealed that 20% of patients with non-metastatic RCC experienced either local recurrence or distant metastases within a 5-year follow-up period after radical treatment of the primary tumors [20]. One study was sourced from the Surveillance, Epidemiology, and End Results (SEER) database, which included 50,815 patients and examined the patterns of distant metastases [21]. In this study, most patients had no metastases at the time of diagnosis, yet half of them did not undergo radical surgery. According to their findings, the lungs were the most common site of solitary metastases (16.8%), followed by lymph nodes (12.1%) and bone (11.1%). In contrast, the liver and brain were uncommon metastatic sites (2.7% and 1.6%, respectively) [21]. Another SEER-based study of 10,410 patients with metastatic RCC detailed the combination of the metastatic sites, which suggested that a pairing of the lung and bone metastases was the most prevalent, representing approximately 10% of all combinations. In contrast, the combination of the bone, brain, and liver was the least common. However, the pattern of recurrence after radical surgery remains uncertain, given that approximately half of this cohort underwent nephrectomies [22]. Fallara et al. conducted a notable study on the recurrence patterns following both open and minimally invasive surgeries (laparoscopic surgery including robotic surgery) in patients with pT1-3aN0 RCC [23]. After matching, each group contained 1019 patients. The number of patients who experienced recurrence was comparable between the two groups (130 patients who underwent open surgery and 125 patients who underwent minimally invasive surgery). Notably, patients who underwent minimally invasive surgery exhibited a higher rate of local recurrence compared to those who had open surgery (hazard ratio [HR]: 2.06, 95% confidential interval [CI]: 1.18–3.58, *p* = 0.01). In addition, compared to open surgery, those who had minimally invasive surgery showed a greater incidence of metastases in atypical sites: areas outside the thoracic and abdominal cavities like the bone, brain, thyroid gland, and skin (HR: 1.09, 95% CI: 1.01–1.16, *p* = 0.04). Identifying the exact cause of this difference is challenging due to the rarity of these metastases. Nonetheless, the potential effects of pneumoperitoneum and the characteristics of tissue manipulation have been suggested as contributing factors.

## 3. Classifications and Definition of Oligo-Metastatic Diseases

Oligo-metastatic diseases are typically defined by the number of metastatic sites, specifically the presence of five or fewer lesions. However, the number of organs involved and the size of the metastatic lesions have not been clearly specified. Treatment strategies and prognoses vary based on the organs involved and the size, number, and growth rate of the metastatic lesions.

Niibe et al. introduced a classification system based on the condition of the primary tumors: oligo-metastasis with uncontrolled primary sites and oligo-recurrence with previously treated primary sites, where all evident recurrence or metastatic sites can be addressed using local therapy [24]. The biological behavior and clinical outcomes of synchronous oligometastatic disease (occurring within a 6-month interval of the primary cancer diagnosis) and metachronous oligometastatic disease (occurring >6 months after the primary cancer diagnosis) differ [24]. Synchronous diseases are typically viewed as more aggressive compared to metachronous ones [17,25]. Previous studies have proposed another definition of “oligo-progression”, which refers to patients who show progression in a few lesions while primary and other metastases are controlled by systemic therapies [17,26,27].

According to the European Society for Radiotherapy and Oncology and the European Organization for Research and Treatment of Cancer, the OligoCare project developed a classification system representing the state of oligo-metastases [17]. First, the imaging-based diagnosis of oligo-metastatic disease was divided into two classifications according to the history of poly-metastatic or oligo-metastatic disease. Patients without a history of poly-metastatic or oligo-metastatic disease were identified as having de novo oligo-metastatic disease. Second, they were divided based on the presence of the primary site and the timing of the occurrence of metastatic disease. Metachronous oligo-metastatic disease, “metachronous oligo-recurrence”, and “metachronous oligo-progression” have been recognized based on the presence of ongoing systemic therapy [17]. Other classifications depend on the history of poly-metastasis, local therapy, or systemic therapy, and these have different treatment goals.

## 4. Novel Imaging to Detect Oligo-Recurrence

Typically, most RCC diagnoses and metastatic sites are based on non-invasive radiological techniques, including ultrasonography (US), computed tomography (CT), and magnetic resonance imaging (MRI). These imaging techniques are the most relevant diagnostic methods for defining oligo-metastatic disease, owing to the scarcity of biomarkers [17]. In terms of nuclear medicine imaging, fluorine-18 fluorodeoxyglucose positron emission tomography/computed tomography ([^18^F] FDG-PET/CT) has been thoroughly examined for diagnosing metastatic RCC [28]. Previous studies have suggested that the expression of glucose transporter 1, which correlates with FDG’s biological activity, varies widely according to the histological subtype of RCC [28]. In contrast, Park et al. found that the accuracy of [^18^F] FDG-PET/CT for diagnosing per-lesion recurrence after radical nephrectomy surpassed that of conventional imaging, including chest and pelvic CT [29]. Several studies have confirmed its effectiveness in detecting the recurrent foci of RCC, although [^18^F] FDG-PET/CT is not deemed a standard imaging test, especially for clear-cell subtypes.

Recently, many studies have highlighted the effectiveness of prostate-specific membrane antigen (PSMA)-based positron emission tomography (PSMA-PET) for detecting oligo-metastatic prostate cancers. PSMA is also found in the tumor-associated neovasculature of primary and metastatic RCC. Several studies have revealed its superiority in identifying metastatic sites compared to conventional imaging and [^18^F] FDG-PET/CT [28]. Moreover, the uptake of PSMA radioligands may serve as an indicator of the efficacy of targeted and immune checkpoint agents [30]. On the other hand, various radiotracers are currently available, each with distinct pharmacokinetic properties. Differences in the diagnostic capacities of RCC are yet to be established. Additionally, to date, no studies have compared their cost-effectiveness with conventional imaging [28]. Further research is required to delineate the role of PSMA-PET/CT in RCC management.

## 5. Current Options in Treating Oligo-Metastatic RCC

The National Comprehensive Cancer Network guidelines suggest MDT, including metastasectomy, SBRT, and ablative therapy, in treating oligometastatic diseases in both stage IV and relapse RCC, regardless of histologic subtypes [31].

### 5.1. Active Surveillance

According to the American Society of Clinical Oncology guidelines, selected patients may be offered initial active surveillance (AS) in addition to MDT [32]. Prospective data on the natural history of metastatic RCC and the safety of AS as an initial strategy are limited. This is because contemporary metastatic RCC trials often excluded patients who did not undergo treatment [33,34]. Nonetheless, AS may be a viable treatment for patients with slow-growing tumors. Current guidelines suggest that AS should be offered to selected patients, specifically those with a limited tumor burden, no symptoms, a favorable histological profile, and a long interval between radical surgery and the development of metastases [35]. Additionally, a previous study hypothesized that some patients with metastatic RCC could safely opt for initial surveillance without compromising their response to later systemic therapies [33]. Although modern medications, including VEGFR-TKIs and ICI drugs, demonstrate high efficacy, they can be toxic and prohibitively expensive for some patients and health systems. Harrison et al. conducted an interesting study in which patient-reported QOL was assessed using completed patient-reported outcomes [34]. The results of these questionnaires indicated that the QOL at baseline was significantly better in patients who underwent AS than in those who underwent systemic therapy. However, it is crucial that clinicians discuss the observation period until disease progression is confirmed.

### 5.2. Surgery

Surgical interventions for metastatic RCC encompass cytoreductive nephrectomy (CN) and metastasectomy. Tumor resection has the potential to be curative if all tumor deposits are removed. In the era of targeted therapy, the advantages of CN are unclear [36]. With the introduction of novel ICI combination therapies, the immediate use of CN in patients with advanced RCC has declined [37]. Furthermore, the role and timing of deferred CN remain controversial. The efficacy of metastasectomies in the treatment of metastatic RCC has also been disputed, given the absence of randomized controlled trials. Existing studies exhibit considerable heterogeneity, primarily because ideal candidates are typically those with a good performance status, prolonged periods without evidence of disease, and relatively minor metastatic burdens [14]. Surgeries for advanced primary and metastatic sites can be invasive, potentially leading to severe complications and delaying or postponing subsequent systemic therapies [38]. Cytoreductive surgery is currently considered an option for patients with fully resectable primary and metastatic lesions. Performing upfront CN for oligo-metastatic RCC is deemed acceptable when complete local treatment can be achieved or when a significant portion of the tumor burden can be excised.

### 5.3. Ablation Therapy

Thermal ablation, such as cryoablation or radiofrequency ablation (RFA), is another option. Ablation is considered a less-invasive treatment option than surgery and has the advantage of repeated treatment. A previous small series showed high local control rates in visceral metastases of RCC [39]. In addition, a previous study suggested that RFA could trigger inflammatory cell activation, similar to radiation therapy [39]. However, the use of thermal ablation for specific metastases may be limited to patients with small single or localized lesions.

### 5.4. Radiation Therapy

Although RCC was previously considered radioresistant, SBRT has emerged as an effective option for oligometastatic RCC [2,40]. A preclinical study examining radiosensitivity in vitro in multiple human cancer cell lines revealed that RCC is relatively resistant to conventional fractionated radiation therapy. Another clinical trial indicated that conventional adjuvant radiation therapy for RCC did not improve local recurrence rates and was associated with severe toxicity [2]. However, radiation therapy, when administered at a higher dose per fraction, has been proven to be effective for RCC. SBRT is characterized as a treatment approach that delivers a high radiation dose to the target using either a single dose or a limited number of fractions, maintaining a high level of precision within the body. Radiation therapy offers a wide range of potential applications across various tumor types and locations, and is both effective and safe. It can also serve as a palliative treatment for lesions that cause local symptoms. Furthermore, the efficacy of SBRT has been demonstrated in unresectable lesions, with high local control rates. The treatment duration is generally brief and reduces patient discomfort owing to its minimally invasive nature. Moreover, in most cases, systemic therapies do not need to be interrupted during the treatment, thus eliminating the risk of tumor progression that could arise from interrupting systemic therapies.

## 6. Treatment Strategy for Metachronous Oligo-Recurrent RCC

In cases of metachronous oligo-recurrence, a particularly favorable subgroup includes patients who manifest metachronous metastases for more than a year after undergoing radical surgery for primary lesions. These patients typically have an indolent disease course. For such patients, any of the approaches, AS, metastasectomy, and SBRT, are viable options. Rini et al. conducted a phase II trial to determine the time to initiation of systemic therapy in patients with metastatic RCC and AS [33]. Of the 48 patients, 47 were systemically treatment-naïve and presented with metachronous, asymptomatic, oligo-recurrent disease. The median duration until the initiation of treatment or withdrawal of consent was 14.9 months. In the multivariate analyses, a greater number of International Metastatic RCC Database Consortium (IMDC) risk factors and an increased number of metastatic disease sites were correlated with a shorter surveillance period. The authors identified a favorable group consisting of patients with either no or one IMDC risk factor and metastases in two or fewer organs, and an unfavorable group comprising all other patients. The former had an estimated median surveillance time of 22.2 months, while the latter’s median was 8.4 months (*p* = 0.0056). Hannan et al. performed a phase II analysis to examine the effect of metastasis-targeted SBRT on outcomes in patients with systemic therapy-naïve oligo-recurrent RCC [41]. They included 23 patients with RCC with up to three extracranial metastases, with a total of 57 sites receiving upfront SBRT. All but one patient had previously undergone radical nephrectomy. Impressively, 91.3% of the patients remained free from systemic therapy for one year. Moreover, the local control rate for sites treated with SBRT stood at 100%. Similarly, Tang et al. published a phase II trial involving patients with oligo-metastatic RCC who had previously undergone nephrectomies [42]. Their findings indicated that SBRT for all metastatic lesions prolonged the disease duration without the need for systemic treatment. Retrospective studies have also supported the efficacy of each approach for oligo-recurrent lesions, identifying several risk factors, such as favorable intermediate-risk IMDC scores and fewer (≤3) metastatic lesions associated with progression [43]. Based on the results of these studies, it is evident that a longer interval since nephrectomy and fewer metastatic sites are associated with improved outcomes.

## 7. Treatment Strategy for Metachronous Oligo-Progressive RCC

The aim of MDT for the treatment of metachronous oligo-progression is to delay the escalation or withdrawal of systemic therapies if alternative lines are limited [44]. When considering disease progression, altering systemic agents should be considered alongside MDT [40]. Nonetheless, previous studies have indicated that combining systemic therapies enhances outcomes and that MDT extending the period before systemic therapy needs to be intensified [45,46]. Among MDTs, radiation therapy may be a more favorable choice than surgical resection when concurrent systemic therapy is used. This preference arises for two main reasons: firstly, radiation can amplify the immune-related effect on the tumor microenvironment, and secondly, there is no need to interrupt systemic therapy during radiation therapy. However, it is crucial to note that exposure to the toxicities of systemic therapies can influence patient OS. Meyer et al. studied a large cohort of 188 patients who underwent SBRT for oligo-metastatic RCC [46]. Their findings revealed that patients with oligo-progression during systemic treatment had a shorter OS compared to those with systemic treatment-naïve oligo-metastatic RCC (23.2 months vs. 33.9 months). De et al. demonstrated that SBRT effectively delayed the intensification of systemic therapy for patients with oligo-progressive RCC, irrespective of the systemic treatment agents used [45]. Notably, half of the patients had not started systemic therapy at the time of progression, and patients receiving TKI at the onset of oligo-progression were at a heightened risk of mortality.

The NIVES study was a single-arm phase II trial designed to evaluate the efficacy of SBRT combined with nivolumab in treating metastatic RCC [47]. Eligible patients had one or more lesions suitable for SBRT and had undergone one or more lines of targeted therapy but were ICI-naïve. The primary endpoint, namely, the objective response rate (ORR), was 17% in this study, which was lower than that observed in the Check Mate 025 trial [48]. This result failed to provide compelling evidence that the combination of nivolumab and SBRT offers additional benefits to previously treated patients with metastatic RCC. Notably, not all metastatic sites in this study were irradiated, leading the authors to suggest that the abscopal effect may be infrequent. In contrast, the SABR-COMET study demonstrated that SBRT at all sites of oligo-metastatic disease led to improved OS compared to palliative radiation therapy across various types of cancer [19]. Because standard systemic therapies for various cancers were permitted in this study, it was challenging to assess the efficacy of SBRT in isolation. Nevertheless, this study implies that it may be essential to treat all metastatic lesions with SBRT, although the effect of radiation beyond the targeted lesion remains uncertain [49].

Several studies have highlighted the efficacy of SBRT in delaying the need for subsequent systemic therapy. Cheung et al. published a phase II study illustrating that SBRT for oligo-metastatic lesions could help delay changing to VEGFR-TKIs [44]. Patients who demonstrated stable responses to systemic therapy for >3 months were enrolled. The study revealed that the cumulative incidence of transition to different systemic therapies was 47% at one year, with a median transition time of 12.6 months. Moreover, the median progression-free survival (PFS) was >9 months, although most cases of disease progression were observed outside the irradiated area. Hannan et al. demonstrated that SBRT prolonged the duration of current systemic therapy in patients experiencing oligo-progression, provided that all lesions were suitable for SBRT [50]. Their findings also indicated that the median time from SBRT to the initiation of new systemic therapy or death was approximately one year. Ma et al. conducted a retrospective study with a small cohort to assess the efficacy of SBRT in the treatment of oligo-progressive RCC using targeted therapies [43]. In this study, favorable and intermediate risk scores in the IMDC risk classification and fewer than three metastases were associated with longer OS and PFS. Thus, SBRT appears beneficial for patients either with stable disease or those with oligo-metastatic lesions that are amenable to SBRT. Consequently, the full irradiation of all metastatic lesions, which are managed by systemic treatments, may offer superior local control for patients with oligo-progressive disease [43].

## 8. MDT with Adjuvant Systemic Therapies for Treatment-Naïve Oligo-Metastatic RCC

### 8.1. SBRT with Adjuvant Systemic Therapies

Numerous retrospective studies have highlighted the efficacy and safety of MDT combined with systemic therapies [51]. Although data regarding the combination of metastasectomy with systemic therapy remains limited [2], SBRT has emerged as a less invasive treatment alternative that may offer prolonged disease-free survival (DFS) owing to its combined systemic effects [2]. This can be attributed to enhanced cancer antigen presentation caused by radiation therapy, which potentially boosts systemic therapy [52,53]. ICIs are promising candidates for combination therapy with SBRT [54,55]. The response rate to ICIs alone is limited to approximately 25% of patients with RCC [56]. However, radiation has been shown to upregulate tumor programmed death ligand 1 (PD-L1) expression, and inhibiting the programmed death 1 (PD-1)/PD-L1 axis using anti-PD-1 drugs enhances antitumor immunity [19].

Siva et al. presented a phase II study illustrating the effectiveness of combining SBRT or conventional RT with a short course of pembrolizumab for the treatment of extracranial oligo-metastatic RCC [57]. They enrolled patients with oligo-metastatic RCC who had received no more than one line of systemic therapy (most patients were systemic therapy-naïve) and those who discontinued their therapies at least one month prior to starting SBRT. On average, pembrolizumab was administrated five days after the final SBRT dose and every three weeks for a total of six months, totaling up to eight cycles. Remarkably, 12 patients (40%) achieved a complete response, with an ORR of 63%. Of those who achieved ORR, 72% exhibited a durable response at the 12-month follow-up. Out of the 41 target lesions treated with SBRT, 15 saw a 100% reduction in size. The cumulative incidence of the first failure at the year mark was 7% for local sites and 30% for distant sites. In terms of progression patterns, most of the patients (12 out of 17) showed distant progression. These results suggest that the optimal combination of temporary ICI with SBRT remains an open question. Additional studies are required to delineate patient selection criteria for adjuvant ICI plus SBRT and to investigate potential adverse events.

### 8.2. Metastasectomy with Adjuvant Systemic Therapies

The effectiveness of perioperative systemic therapy for treating RCC with metastases remains unclear. Although several randomized controlled studies have attempted to demonstrate the value of adjuvant therapies ranging from cytokines to chemotherapy for high-risk RCC, only two studies achieved the primary endpoints [58]. A phase III study, the IMmotion010 trial, did not show that adjuvant therapy enhances the prognosis of high-risk or oligo-metastatic RCC [59]. Participants were assigned to receive either atezolizumab or a placebo after undergoing nephrectomy, with or without metastasectomy, and had no evidence of disease after surgery. Notably, both groups had a small percentage (14% and 13%, respectively) of patients who underwent metastasectomy, and there was no observed difference in DFS between the two groups. In contrast, the Keynote 564 trial enrolled RCC patients with high-risk factors for recurrence [60]. These patients underwent nephrectomy, with or without metastasectomy, and any metastases were completely removed. Inclusion criteria considered patients with high-risk factors based on pathological features, the presence of regional lymph node metastases, and stage M1, but no evidence of disease following metastasectomy, were included. Pembrolizumab was associated with a significantly longer DFS than the placebo (DFS at 24 months: 77.3% vs. 68.1%, *p* = 0.002). Yet, the median OS was not reached in either group (HR for death: 0.54). The efficacy of ICI as adjuvant therapy remains controversial, especially because these trials included only a limited number of patients who had undergone surgical resection of distant metastases. Moreover, the optimal duration of adjuvant therapy remains uncertain, even though the trial set it at one year.

## 9. Brain and Bone Metastases

Among the organs involved, brain metastases often present with poor prognoses, even when appropriate local treatments are administered. However, local treatment is essential for brain metastases [61]. This is primarily because the blood–brain barrier consistently challenges the treatment of RCC brain metastasis despite advancements in systemic therapies [2]. Similarly, bone metastases are recognized as indicators of poor prognosis [62]. Local therapy is crucial for preventing or alleviating neurological symptoms and pain. According to current guidelines, local treatment for both brain and bone metastases is recommended in all cases [32]. Conversely, many randomized controlled studies often exclude brain and bone metastases in RCC because evaluation using the Response Evaluation Criteria in Solid Tumors (RECIST) poses challenges. To date, only a few retrospective studies have assessed the efficacy of local therapies for such metastases. Franzese et al. presented a risk classification for patients with primary kidney cancer and oligo-metastases who were treated with SBRT for both intracranial and extra-cranial metastases [63]. Notably, approximately half of the patients underwent systemic therapy before SBRT. The presence of brain metastases and a history of treatment for bone metastases were significant determinants of OS. Remarkably, patients with brain metastases had a 3-year OS of 9.7%, in stark contrast to 82.6% of those without either bone or brain metastases. Hence, determining the optimal combination of systemic and focal therapies is vital for these patients. Other studies have provided interesting information on multidisciplinary treatments. Huntoon et al. demonstrated the superior efficacy of combined surgery and radiation therapy with systemic therapy compared with that of combining radiation therapy with systemic therapy, specifically for brain metastases [64]. Navarria et al. found that combining surgery with multifractional radiosurgery yielded better outcomes than single or hypofractionated radiation therapy [65].

With respect to bone metastases, Liu et al. highlighted that oligo-metastatic bone disease exhibited a more favorable prognosis compared to non-oligo-metastatic bone metastasis when treated with sunitinib [66]. Onal et al. conducted a study on RCC patients with only bone oligo-metastasis who were treated with SBRT [67]. They revealed that patients who underwent treatment for spinal metastases had a better OS than those with non-spinal metastases. Furthermore, patients who received a single SBRT fraction demonstrated a better PFS than those who received multiple SBRT fractions. This suggests that a higher dose of SBRT administered in fewer fractions may be more effective for treating bone metastases.

## 10. Conclusions and Future Directions

Managing oligo-metastases presents a promising approach for patients with oligo-recurrence with a favorable risk profile and is expected to have better outcomes. More aggressive forms require a comprehensive approach involving both systemic and local therapies. Among MDT options, SBRT is a well-established and emerging treatment choice, offering potential curative or palliative benefits. Its advantages include its minimally invasive nature and the absence of interruptions to systemic treatments (Figure 1 and Figure 2). SBRT is effective when combined with controlled systemic therapy for oligo-progressive RCC, potentially delaying the escalation of such treatments in specific patients. This underscores the importance of managing the overall aggressiveness of the disease, although radiation therapy may influence metastasis throughout the body owing to its effects on the tumor microenvironment and areas beyond the irradiated sites. When combined with SBRT, adjuvant ICI therapy has shown promising oncological results for oligo-progressive RCC. However, the optimal treatment regimen and duration remain unclear. Further investigation is imperative to ascertain the combined efficacy of MDT and ICIs and to determine the ideal treatment duration.

## Figures and Tables

**Figure 1 cancers-15-05873-f001:**
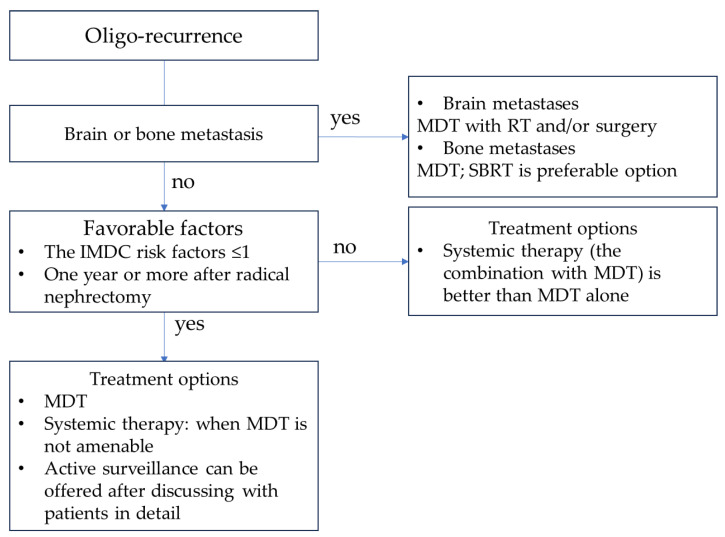
Treatment strategy for oligo-recurrent RCC. Abbreviations: MDT, metastasis-directed therapy; RT, radiation therapy; SBRT, stereotactic body radiation therapy; IMDC, International Metastatic RCC Disease Consortium.

**Figure 2 cancers-15-05873-f002:**
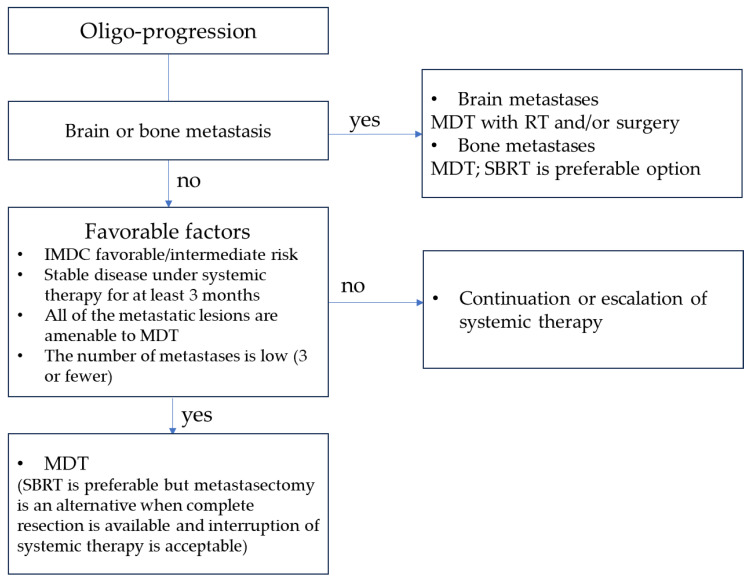
Treatment strategy for oligo-progressive RCC. Abbreviations: MDT, metastasis-directed therapy; RT, radiation therapy; SBRT, stereotactic body radiation therapy; IMDC, International Metastatic RCC Disease Consortium.

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
