# Peer review of "The Diagnosis and Treatment Approach for Oligo-Recurrent and Oligo-Progressive Renal Cell Carcinoma"

_cancers, 2023, doi:10.3390/cancers15245873_

Round 1
Reviewer 1 Report
Comments and Suggestions for Authors
Summary:
The manuscript provides a comprehensive and well-structured review of the evolving strategies in managing oligorecurrence and oligoprogression in renal cell carcinoma (RCC). The authors successfully navigate through the complexities of metastasis-directed therapy (MDT), specifically focusing on the role of stereotactic body radiation therapy (SBRT) and immune checkpoint inhibitors (ICIs). The integration of these modalities and their potential synergies are thoroughly discussed.
Comments:
1. The subsections and the details covered are adequate and detailed.
2. The paper presents the literature and concepts clearly.
3. The paper addresses an intriguing topic and presents valuable insights into diagnosis and treatment approaches.
4. The paper is well-written, and the results are promising.
5. The figures are good but it will be good if more figures can be added to aid the content.
Author Response
We sincerely appreciate the reviewer’s constructive comments. We concur with the reviewer’s suggestion that incorporating additional figures could enhance the clarity of the manuscript. Regrettably, we were unable to identify suitable figures to include, and the tables delineating the outcomes of the studies analyzing the efficacy of SBRT for oligo-metastatic RCC have already been published.
Reference; Michael Christensen and Raquibul Hannan. The Emerging Role of Radiation Therapy in Renal Cell Carcinoma, Cancers 2022, 14, 4693, https://doi/10.3390/cancers14194693
Reviewer 2 Report
Comments and Suggestions for Authors
The review is well-executed, thoroughly examining recent publications, presenting the topics clearly, and maintaining accessibility. The concluding images summarizing the management of oligorecurrent metastasis and oligoprogression are informative, though there’s room for improvement in terms of graphic design.
Author Response
We would like to thank the reviewer for the helpful comments. We have revised the figures to improve the appearance.
Reviewer 3 Report
Comments and Suggestions for Authors
This manuscript entitled “The Diagnosis and Treatment Approach for Oligorecurrent and Oligoprogressive Renal Cell Carcinoma” provides a comprehensive and detailed overview of the current evidence related to the management of oligo-recurrent and progressive renal cell carcinoma (RCC). It effectively discusses the concepts of oligorecurrence and oligoprogression in oncology, the primary strategy for recurrence and metastases in RCC, and the potential of metastasis-directed treatments (MDTs) such as stereotactic body radiation therapy (SBRT).
Authors explored the possibility of combining MDT with immune checkpoint inhibitors (ICIs) to yield better results owing to their synergistic effects. Moreover, authors discussed the classifications and definition of oligo-metastatic diseases, the novel imaging techniques to detect oligo-recurrence, the current options in treating oligo-metastatic RCC, the treatment strategy for metachronous oligorecurrent and oligoprogressive RCC, and the treatment approaches for brain and bone metastases in RCC. This is a significant contribution to the field and could pave the way for future research.
However, it would be beneficial if the authors could provide more detailed information on the selection criteria for patients who would benefit from this combined approach. Additionally, a discussion on the potential side effects and the management of these side effects would also be valuable for the readers.
Finally, the manuscript is well-written and provides valuable insights into the management of oligo-recurrent and progressive RCC. It is a worthy contribution to the field and should be considered for publication in Cancers.
Author Response
We extend our appreciation to the reviewer for their insightful comments. We acknowledge the importance of specifying suitable patients in the combined treatment of ICI and MDT. However, it is crucial to note that the treatment's efficacy was demonstrated solely in a clinical trial with a limited number of participants. Identifying both patient selection criteria and potential adverse events based on the study results might pose challenges. Therefore, we have stated that additional studies are required to delineate patient selection criteria for adjuvant ICI plus SBRT and to investigate potential adverse events and added the sentence to the text (page 7, lines 48-49).